# Single rate-limiting event of carcinogenesis

Yutaka Yasui [1,2] ✉ & Qi Liu [2]

Single-cell studies have discovered abundant cancer-associated genetic/phenotypic changes in non-cancerous cells, strikingly contrasting with the infrequency of cancer. Epidemiological data have revealed decades-long plateaus of breast cancer incidence in the contralateral breast and twins/relatives following the first/proband's diagnosis, unlike the well-known continuous increase of population-level incidence with age, the latter ostensibly attributable to the successive accumulation of multiple genetic/epigenetic changes necessary for transformation. Here, we explain these contradicting observations by differentiating cell-level, individual-level, and population-level evidence. First, we show the same decades-long incidence plateau for renal-cell carcinoma in the contralateral kidney following the first diagnosis, expanding the individual-level evidence from breast cancer. We then consider somatic evolution and cell competition in stem-cell compartments and their bounded nature in ageing as a hypothesized mechanism for the abundant cancer-associated cell-level changes and the prolonged constancy of individual-level incidence. Individual-specific propensity with heritable/familial components underlies this process, with which we show congruence between individual-level's constant incidence vs. population-level's increasing incidence. The resulting postulate, an extension of one by Peto and Mack 25 years ago, distinguishes the last multistage "hit" as the critically rate-limiting event in carcinogenesis. Its supporting evidence calls for a reappraisal of the precise nature of multistage carcinogenesis in cancer biology.

A large body of evidence supports the multistage carcinogenesis theory: for a transformation of a normal cell to occur, it must accumulate several genetic and epigenetic changes (hereafter loosely referred to as "driver mutations"), which enables cells to escape various inbuilt failsafe mechanisms of tumor suppression. At the molecular level, these changes are characterized as 6–8 essential alterations of transformed cells, often referred to as the hallmarks of cancer[1–3]. Accumulating a specific set of driver mutations successively requires time: the chance for a cell to acquire a necessary set for transformation must increase monotonically with time. At the population level, indeed, cancer incidence rates increase as a power of age for most cancer types[4], triangulating the support for the theory. Specifically, population scientists have observed as early as in the 1950s that population-level cancer mortality (and incidence) rates increase approximately as the 6th power of age, which on a log–log plot is approximately linear

with a slope of 6 (i.e., power-law increase)[5,6]. They depicted this observation with a stochastic-process model of carcinogenesis, often referred to as Armitage-Doll Model, endorsing that if transformation of a cell requires $k$ successive driver mutations, the incidence rate increases as the ($k$-1)th power of age, empirically implying that 7 successive driver mutations may be required for a normal cell to transform.

Two lines of evidence call for further appraisal of the nature of multistage carcinogenesis, however. One from biological science is the series of sequencing studies at the single-cell level showing that cancer-associated genetic changes, sometimes reported as field cancerization, are widespread in non-cancerous cells in various tissues, especially in ageing[7–10]. In addition to genetic changes, highly prevalent pre-cancerous phenotypic changes have been reported with increasing spatial and genetic resolutions[11–14]. For example, a recent

[1]St. Jude Children's Research Hospital, Memphis, TN, USA. [2]University of Alberta, Edmonton, AB, Canada. ✉e-mail: yutaka.yasui@stjude.org

machine-learning-based 3D histological study on the microanatomy of pancreas at the single-cell resolution identified an average of 13 pancreatic intraepithelial neoplasia, a precursor to pancreatic cancer, per cubic centimeters of the pancreas, each with a distinct somatic mutation profile, which extrapolates to hundreds of pancreatic intraepithelial neoplasia in an adult pancreas[14]. Cancer is a considerably rarer event compared to the frequencies of these cancer-associated genetic/phenotypic changes, implying that there is a critically rate-limiting step following these abundant changes prior to transformation.

The other piece of evidence, one from population science, is the epidemiological data on breast cancer incidence that seemingly contradicts with both the theory and its supporting evidence of the power-law increase in population-level cancer incidence with age. The rates at which breast cancer develops in the contralateral breast and in female relatives, including twins, after a woman's diagnosis with the disease remain constant and high for decades, irrespective of the age at the proband's diagnosis[15–18]. Based on these observations, Peto and Mack postulated: "incidence in susceptible women increases to a high constant level by a predetermined age that varies between families", but recognizing "this seems inconsistent with conventional models of carcinogenesis and susceptibility"[15]. If an accumulation of 6–8 driver mutations is required for transformation, which takes time, why doesn't breast cancer incidence in the contralateral breast and the relatives of patients increase over time? How can their incidence remain constant when the population-level incidence increases as a power of age?

A recent large population-based cohort study from England also reported data that show women with non-screening-detected ductal carcinoma in situ (DCIS) had a steady high incidence of invasive breast cancer over 25 years after DCIS treatment, again largely irrespective of the age at DCIS diagnosis[19]. Note that, according to the stochastic-process theory of the multistage carcinogenesis above, the incidence invariant over time corresponds to a state of $k = 1$ from which only a single stochastic event is required for transformation: we refer this state of $k = 1$ as "precancer state" and cells in the precancer state as "precancer cells" hereafter. These sound epidemiological data, therefore, imply that the number of precancer cells is stable in the contralateral breast and female relatives' breasts after a woman's breast cancer diagnosis and in women's breasts after non-screening detected DCIS.

There are other pieces of epidemiological evidence for the constant incidence rates of cancer. For example, lung cancer incidence rises as the 4th power of the duration of smoking exposure, then remains roughly constant when smoking stops[20]. Incidence rates of cervical cancer, which is influenced by HPV infection, remain roughly constant within birth cohorts after middle age when active infection is much rarer[21].

The postulate of Peto and Mack above ("Peto-Mack Postulate") has a conceptual significance on carcinogenesis mechanisms as it specifies the form of *individual-level* cancer risk by age. How cancer risk of an individual changes with age cannot be observed directly because most individuals do not develop cancer, some develop only one cancer, and in a small subgroup with multiple primary cancers, subsequent cancer risk may be influenced by the preceding cancer treatment. Thus, the individual-level age-risk pattern has been inferred from the population-level age-incidence patterns instead; that is, individual-level risk has been considered to increase as a power of age. Peto and Mack's work challenged this inferred individual-level age-risk pattern from the population-level data and provided evidence for a very different pattern: cancer risk of an individual comprises two phases, the first phase where the risk increases with age, followed by the second phase where the risk plateaus. If the cancer risk of an individual is constant for a prolonged period after the initial increase with age, it has significant implications on how cancer arises in an

individual and calls for research to assess and exploit this conjecture for advancing cancer biology and prevention strategies.

Possibly due to its inconsistency with conventional models of carcinogenesis and susceptibility, Peto-Mack Postulate has not been furthered since their publication 25 years ago, in spite of its strong epidemiological evidence[15]. Here, we shed light on the Postulate's individual-level age-risk pattern, augmenting with somatic evolution and cell competition as its potential underlying cell-level process. The augmentation gives a hypothesized explanation on the two-phase form of incidence and ties the Postulate to the widespread cancer-associated genetic/phenotypic changes. We also resolve the key contradiction of the constant individual-level age-risk pattern of the Postulate with the well-documented power-law increase of population-level cancer incidence with age. For the resolution, we explicitly consider subdivisions of a population into subpopulations with varying levels of "propensity" which represents the force of cancer development determined by individuals' genetic and environmental/lifestyle factors, and their complex interplay, with heritable/familial components. The resolution analysis of various cancer types reveals the propensity subpopulations whose frequencies are consistent with both the notion of selective pressure and the known early-life risk for selected cancer types, further attesting the plausibility of the Postulate.

## Results

### Second cancer in the contralateral organ

Cancer incidence data in the contralateral organ following the first diagnosis offers valuable clues on individual-level cancer risk over time. Breast cancer is by far the most studied cancer in this regard due to its relatively high incidence. We asked whether the decades-long constant incidence in the contralateral organ after the first diagnosis is a phenomenon specific to breast cancer or a more general phenomenon. Figure 1 shows the renal-cell carcinoma incidence in the contralateral kidney according to years after the first diagnosis, based on the US SEER cancer registry data, overall and by the age at the first diagnosis[4]. The cumulative hazard rates increase strictly linearly with time over a few decades following the first diagnosis, indicating that the incidence rates are remarkably constant. The prolonged constancy of incidence is seen irrespective of the age at the first diagnosis, consistent with the findings on breast cancer. The plots for older ages at the first diagnosis show influence of mortality risk in later years, presumably correlated with the risk of renal-cell carcinoma, thereby bending the curves towards lower incidence, and/or due to small numbers of subjects at risk ($n = 538$ at 25 years and $n = 498$ at 20 years for the 1st diagnosis in 60–69 and 70+ years olds, respectively): nevertheless, they do show linearly-increasing lines for a decade or two following the first diagnosis.

### Somatic evolution/cell competition effect

We consider somatic evolution and cell competition as the potential cell-level processes underlying the two phases of cancer risk at the individual level. Accumulation of somatic mutations over time gives stem cells varying fitness advantages, driving cell competition among them, resulting in expansions of higher fitness clones and removal of the others[7–9,22,23]. However, this process does not appear to persist throughout lifetime. As shown by Mitchell et al.[22] for hematopoiesis and reviewed by van Neerven and Vermeulen[23] for various tissues, emerging evidence indicates a drastic shift in somatic evolution and cell competition with ageing, marked by a significant and sudden reduction in clonal diversity with increased individual clone sizes in stem-cell compartments. This sudden shift has also been demonstrated in an elegant mechanistic-modeling simulation of cell competition over ageing, using approximate Bayesian computation[22]. Consequently, a stem-cell compartment in ageing comprises a small

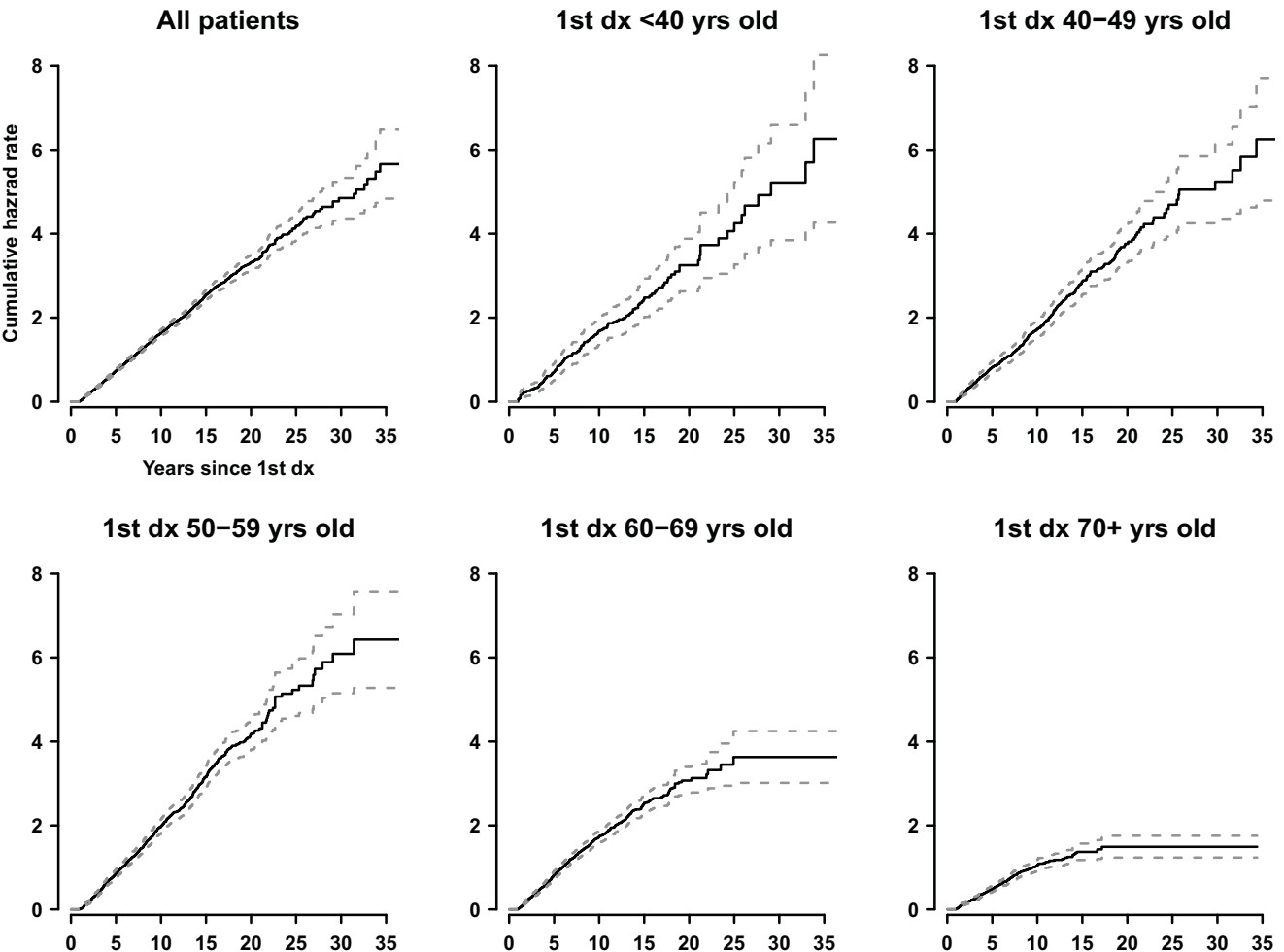

**Fig. 1 | Renal-cell carcinoma incidence in the contralateral kidney following the first renal-cell carcinoma, overall and by age at the first diagnosis, based on the Surveillance, Epidemiology, and End Results cancer registry data of the United States, including the diagnoses of 1975–2021.** The plots show estimated cumulative hazard rates by years since the first diagnosis. A linear increase of the cumulative hazard rates over time indicates a constant incidence rate, which is seen in all panels.

number of clones, or even a single clone ("fixation"), reaching a static state with respect to the composition of the compartment's cell population. For example, Mitchell et al. reported a reduction of 3–4 orders of magnitude in the number of stem cells contributing to hematopoiesis, from 20,000 to 200,000 to less than 20, by 75 years of age[22]. This bounded nature of somatic evolution/cell competition in ageing, whether it is due to spatial limitations[22–24] or ageing-related factors[22,23], resonates with the two phases of individual-level cancer risk and the increase-to-plateau transition of Peto-Mack Postulate. Phase 1 is the period of somatic evolution/cell-competition when driver mutations are stochastically acquired and accumulated in the diverse stem cells, and, through cell competition, the number of higher-fitness stem cells, including very rare ones that entered the precancer state, increases as a power of age. Phase 2 is the period of a static state of the stem-cell population with little diversity, which commences when the stem-cell compartment gets dominated by a small number of, or a single, high-fitness clone(s) that are large in size: thus, the number of cells at risk for transformation (i.e., precancer cells) remains unchanged in Phase 2. Figure 2 depicts this mechanism, which explains not only the transition from the first phase to the second, but the shapes of their age-specific rates. The number of precancer cells increases as a power of age in Phase 1 (a straight-line increase in a log-log plot equals a power-law increase) when driver mutations are accumulated and high-fitness cells expand greatly, while

it does not change with time in Phase 2: the cancer risk of an individual tracks the number of precancer cells and thus increases in the former but remains unchanged in the latter.

## Peto-Mack postulate of carcinogenesis
Peto and Mack's postulate for breast cancer incidence was: "incidence in susceptible women increases to a high constant level by a predetermined age that varies between families"[15]. Here, we explicitly write it in an equation and consider it as a model of individual-level cancer risk (not just that of breast cancer), consistent with the cell-level processes in Fig. 2 above:

$$\text{Individual--level Incidence Rate}_i = \begin{cases} \text{Phase 1}: & P_i' \times Age^{k-1} & \text{if } Age < a(P_i') \\ \text{Phase 2}: & C' & \text{if } Age \geq a(P_i') \end{cases}$$

(1)

where $P_i'$ represents the individual propensity of the $i$-th individual, $(k-1)$ is the exponent of the power-law increase with age in Phase 1, $C'$ is the high constant incidence rate in Phase 2, and $a(P_i')$ is the predetermined age of the $i$-th individual at the transition from Phase 1 to Phase 2 (Fig. 3).

Note that the individual's propensity $P_i'$ that has heritable/familial components, which Peto and Mack described as "varies between families", determines both the incidence rate of any age and the age of

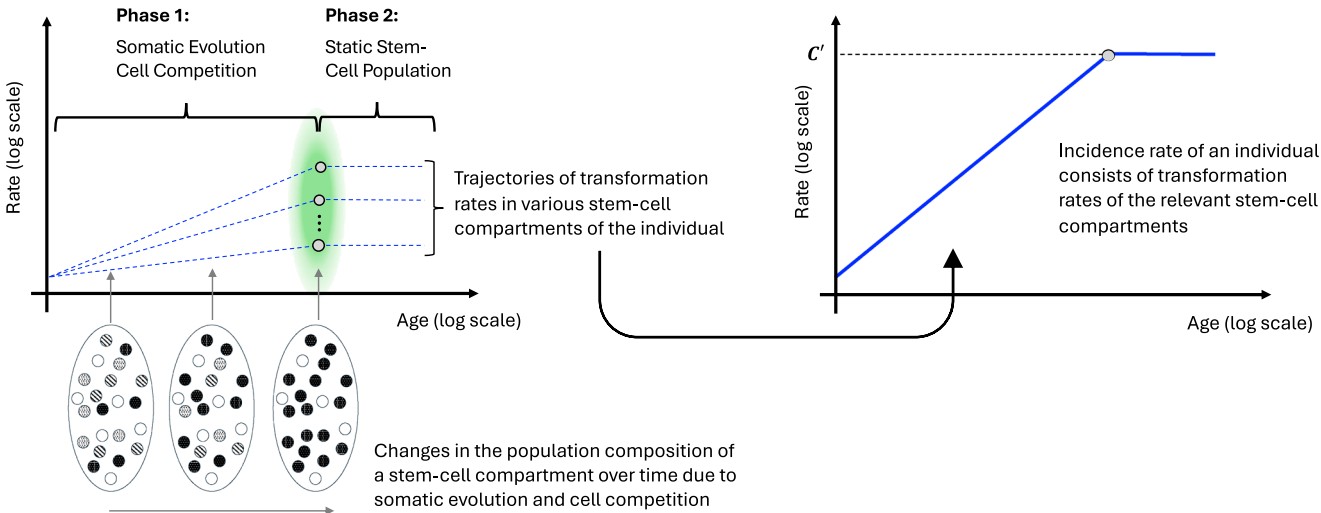

**Fig. 2 | From cell-level transformation rates to individual-level cancer risk.** An illustration of changes in the population composition of a stem-cell compartment due to somatic evolution and cell competition and the corresponding trajectory of transformation rates in the stem-cell compartments. Stem cells acquire various somatic mutations over lifetime which, through cell competition, results in a static population of the stem-cell compartment in ageing, with high-fitness clones of large sizes (black and white cells) occupying the compartment. This leads to the constant transformation rate of each stem-cell compartment. An individual's incidence rate of a particular cancer type is determined by the transformation rates of stem-cell compartments of relevant tissues.

the phase transition, $a(P'_i)$, in Model (1). The inconsistency stated by Peto and Mack with the conventional models lies in Phase 2: without the plateauing of risk in Phase 2, the risk in Phase 1 increases as a power of age, which agrees with Armitage-Doll Model (see Methods) and the requirement of accumulation of multiple driver mutations for a cell to transform.

**Empirical fit of the Postulate model**

How can the age-invariant risk of Peto-Mack Postulate be consistent with the population-level incidence rates of various cancer types that increase as a power of age? To address this question, we considered the Postulate model (1) with up to six subpopulations with distinct individual-level propensities and assessed its goodness of fit to the population-level age-specific incidence rates in 5-year age groups of various cancer types: see Methods for details. Note that between-individual heterogeneity in propensity, as represented in the Postulate model (1), exists on a continuum due to genetic and environmental/lifestyle risk factors[25,26] and extends beyond the discrete subpopulations, but we used the six subpopulations for modeling convenience here. Figure 4 shows Model (1)'s fits to the population-level pancreatic cancer incidence in females as an example. The red line shows the approximately linear increase of the population-level incidence rates with age in the log-log plot, indicating the power-law increase with age in the original scale. Panel (A)'s analysis considered $k = 7$ in the Postulate model (1) prior to the plateau, consistent with the Nordling's[5] and Armitage-Doll Model[6]. The best number of subpopulations with varying propensities, selected using Akaike's Information Criterion (AIC)[27], was three, which reach a high constant incidence rate of 0.5% per year at 65, 75, and 85 years old, with their proportions in the population of 2.57%, 5.70%, and 11.42%, respectively. Even with the fixed $k = 7$ and only 3 subpopulations, the Postulate model (1) produced a very good fit to the 11 age-specific population-level incidence rates (Fig. 4, Panel (A)).

Panel (B)'s analysis considered the identical setting as Panel (A)'s, with the only difference being the high constant rate of 1.0% per year

instead of 0.5% per year. The best fit is identical between the two analyses of Panels (A) and (B), with the only difference being the subpopulation proportions, which became exactly half in Panel (B)'s analysis. This demonstrates that the high constant rate in Peto-Mack Postulate is not an estimable parameter in our analytic framework from the population-level age-specific incidence rates (note, however, that the constant rate is observed directly in the contralateral breast cancer rate[15]). Analyses of Panels (C) and (D) considered the identical setting as that of Panel (B) with the only difference being the value of $k$, $k = 10$ and $k = 13$, respectively, instead of $k = 7$. The best AIC models had the 6 subpopulations in both analyses, with Panel (C)'s $k = 10$ model showing a slightly better fit than the models of Panels (B) and (D). This indicates that similar fits can be obtained for a range of $k$ in Peto-Mack Model, and there is little evidence that $k = 7$, the value suggested in the original analysis of multistage carcinogenesis modeling by Nordling[5] and Armitage and Doll[6], is supported more than other values by the population-level age-incidence data. Note also that, under the best fit model of Panel (C), the subpopulations that reached the high constant rate at ages before 45 years old were quite small (0.07% of the population), compared to those reaching the high constant rate at 45 or older (9.79% of the population, 134 times larger), consistent with the notion that inherited components of high propensity are subject to selective pressure. Furthermore, the majority of the population seems to have propensity that is low enough to not reach the high constant rate during their lifetime, consistent with the point made by Peto and Mack for breast cancer. These features are consistent across all panels, and their agreement across models with different parameterizations supports their biological plausibility, mitigating concerns about overfitting artifacts of the modeling.

**Empirical fit to various cancer types**

The same analyses of Panels (B–D) of the female pancreatic cancer incidence above, with an addition of $k = 4$, which replaced Panel (A), were conducted with each of the following cancer types, for males and females separately, using the observed population-level age-specific

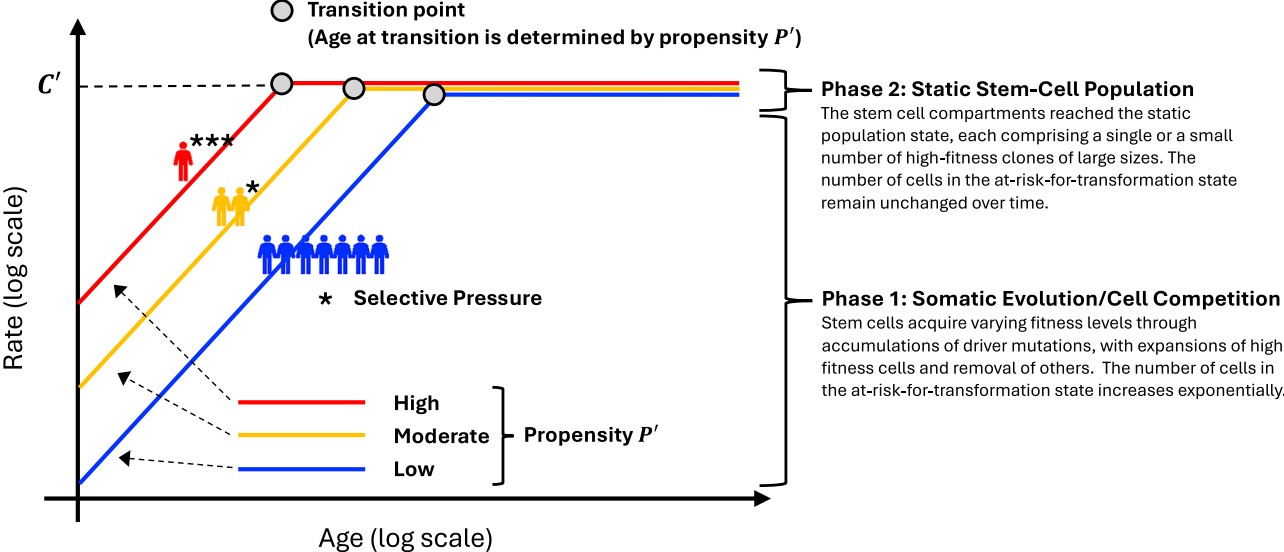

**Fig. 3 | A graphical illustration of Peto-Mack Postulate and its two phases of individual-level cancer risk in the log-log scale, augmented with somatic evolution/cell competition, with three subpopulations of high, moderate, and low levels of propensity $P'$.** Individuals with higher propensity reach the transition point from the increasing incidence (Phase 1) to high constant incidence $C'$ (Phase 2) at younger ages: the heritable/familial components of propensity are subject to selective pressure, leading to lower prevalence of higher propensity individuals.

incidence rates in 5-year age groups from the SEER populations: brain; gallbladder; kidney; leukemia; melanoma of skin; non Hodgkin lymphoma (NHL); pancreas; stomach; and female breast. A summary of these fits is given in Table 1: Supplementary Figs. S2 and S3 provide the corresponding figure for each cancer type by sex; and Supplementary Fig. S4 shows the corresponding figure for female breast cancer.

Although incidence and its age-associated patterns vary greatly across cancer types and sexes, Table 1 shows similar features across all cancer types and sexes as those of female pancreatic cancer: (i) little evidence supporting $k = 7$ over $k = 10$ or 13 (the 2nd column of Table 1); and (ii) the subpopulations with higher propensities that reach the high constant rate before age 45 years old are consistently small, accounting for <1% for all cancer types in both females and males, except female melanoma of skin and breast cancer (the 3rd column of Table 1), and the subpopulations with lower propensities that reach the high constant rate at or after 45 years old are substantially larger (the 4th column of Table 1). The feature ii) seen across cancer types and sexes is consistent with selective pressure of the inherited components of higher propensities. On the other hand, for brain cancer, leukemia, melanoma of skin, and NHL, the subpopulation with the highest propensity was estimated to be larger than the subpopulation with the second highest propensity, and this was seen invariably in both sexes (the 5th column of Table 1). This feature is not recognizable by visual inspection of the population-level incidence rates. These are the four of the eight cancer types we examined that appear more frequently in the first few decades of life, including childhood, implying the existence of a small subpopulation with very high propensity of these cancer types that lead to early onset.

## Discussion

Mathematical modeling of population-level age-specific cancer incidence has a distinguished history. Nordling[5] and Armitage and Doll[6] demonstrated that population-level mortality (incidence) rates of many types of cancer rise as the 6th power of age, implying seven rate-limiting steps. Moolgavkar and Knudson[28] proposed an elegant clonal-expansion model with only two stages, estimating parameters representing cell kinetics of normal and intermediate cells. Pike et al.[29] and Rosner and Colditz[30] advanced the Armitage-Doll framework for breast

cancer by introducing "breast tissue age", reflecting a life-course sequence of hormonal and reproductive events known to modify the risk. Despite their conceptual differences, these landmark models all fit population-level incidence well. Our work continues this tradition while offering a fundamentally different perspective, the Peto-Mack Postulate (Fig. 3), which Peto and Mack formulated from breast cancer findings[15] unavailable to earlier models. After further supporting the Postulate with contralateral organ incidence data (DCIS and renal cell carcinoma), we showed that the Postulate's two-phase structure, i.e., an initial Armitage-Doll-like rise in incidence as a power of age, followed by a sustained high plateau, provides robust fit to population-level age-specific incidence rates across a variety of cancer types (Table 1). This fit was enabled by individual "propensity", which has familial and hereditary components and determines both the individual's age-dependent rate (rising as a power of age) and the age at which the plateau begins. Given the constant incidence, which signifies a state requiring only of a single stochastic "hit" for malignant transformation[5,6], and the abundance of precancer cells relative the rarity of cancer[7-14], we reasoned that the final step for transforming from precancer to cancer represents the most biologically restrictive step in the multistage carcinogenesis process, i.e., the critically rate-limiting event.

There are a few important implications of our augmentation to Peto-Mack Postulate involving somatic evolution and cell competition. The first implication is on the diversity of driver mutations and independence of the phase-transition age from the specific sets of driver mutations stem cells acquire. Because each stem cell acquires a set of somatic mutations stochastically, different stem-cell compartments would get dominated by different high-fitness clones at phase transition. This is consistent, for example, with distinct somatic mutation profiles of numerous pancreatic intraepithelial neoplasia found in adult pancreas[14]. For the contralateral breast and female twins/relatives to have a high constant incidence following a woman's breast cancer diagnosis (and the similar phenomenon occurs for renal-cell carcinoma), however, stem-cell compartments must reach the phase transition approximately at the same age, not only within an individual in the contralateral organ, but among twins/relatives. For this to occur, the specific sets of somatic mutations acquired in various stem cells

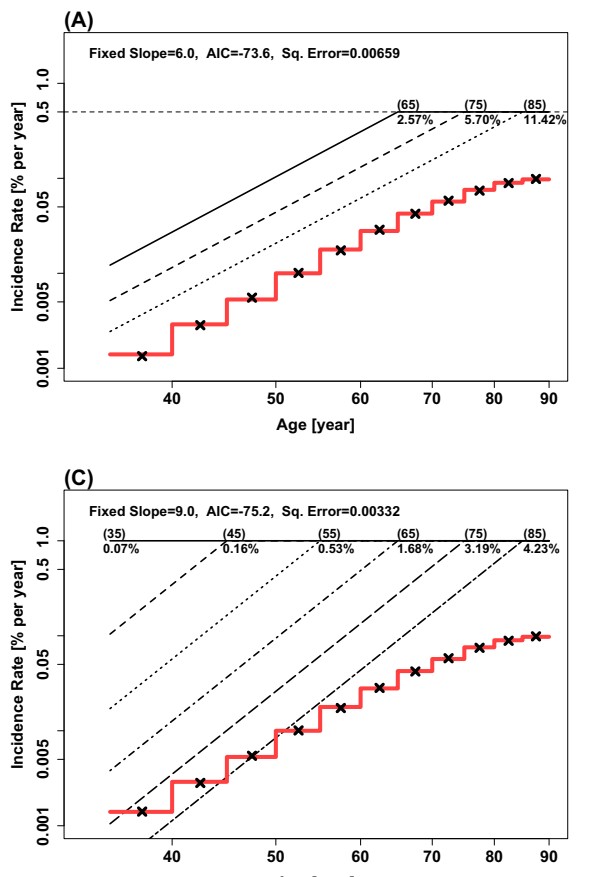

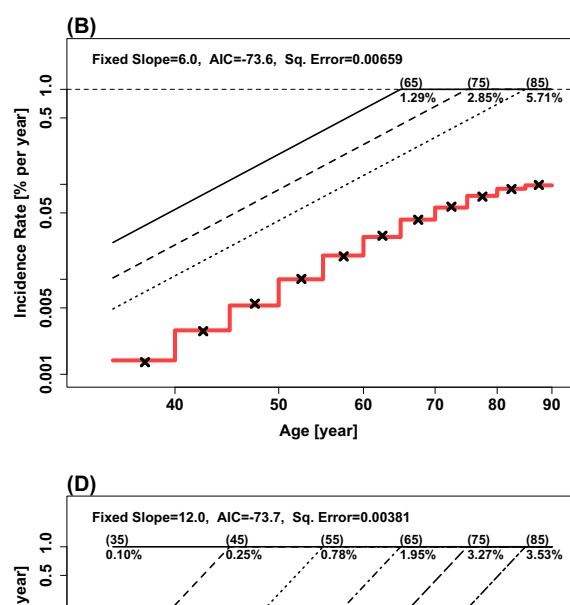

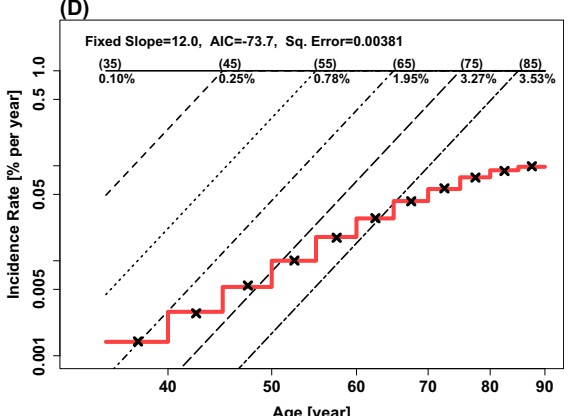

**Fig. 4 | Fits of Peto-Mack postulate model with up to six subpopulations with distinct propensity levels to the observed population-level age-specific incidence rates in 5-year age groups of pancreatic cancer in females from the Surveillance, Epidemiology, and End Results program.** All panels are in the log-log scale. Panels **A**, **B** considered $k = 7$, while Panels **C**, **D** considered $k = 10$ and $k = 13$, respectively. The high constant rate of the model was set at 0.5%/year in (**A**), and 1.0%/year in (**B**, **C**, **D**). In each analysis, the subpopulation size, $m$, of 6, 5, 4, and 3 were considered where $m = 6$ subpopulations are those reaching at the high constant rate at age 35, 45, 55, 65, 75, and 85, and $m < 6$ subpopulations would remove the first (6-m) of these subpopulations. The fits were obtained by least square methods (minimizing the sum of squared errors), and the best subpopulation size, $m$, can be selected by Akaike's Information Criterion (AIC).

must have little influence on the time to, and age at, the transition, given propensity $P'_i$ (and the propensity's heritable/familial component must be large). While specific sets of driver mutations acquired differ, the relative distribution of fitness acquired by stem cells in a stem-cell compartment and its evolution over time must be similar across stem-cell compartments of an individual and of individuals with similar propensity levels. In other words, there must be greatly diverse sets of driver mutations with which stem cells could acquire similarly high (or low) fitness levels, and this diversity makes the fitness distribution of the stem cells similar across stem-cell compartments, given the similar propensity, in spite of the stochastic nature of driver-mutation acquisition and the dynamic nature of cell competition among clones (possibly including malignant clones[31]). The time to the phase transition must, therefore, be largely determined by propensity $P'_i$ and not by the specific sets of mutations stem cells acquire.

The second implication is on the familial nature of propensity $P'_i$. For propensity $P'_i$ to determine the age at phase transition that "varies between families", the factors that determine propensity $P'_i$ must be strongly correlated within families. That is, inherited germline mutations (instead of de novo germline mutations) and environmental/lifestyle factors that are shared strongly among family members must influence propensity $P'_i$. Note that non-heritable/non-familial factors can also influence the risk: as illustrated by Fig. S1 Panel (B), between-individual variation around the mean incidence of the subpopulation can exist and permitted in the Postulate model.

The third implication, and perhaps the most important one for cancer prevention, is that the majority of cancer develops in Phase 2 where the final, rate-limiting "hit" occurs at a substantially higher and constant rate compared to Phase 1. It implies that identifying and preventing this final "hit" is crucial for developing effective cancer prevention strategies, as it seems to be the rate-limiting step of carcinogenesis. The individual's propensity may also be altered for lowering risk, but its familial nature suggests a large hereditary component that may be difficult to intervene. Furthermore, many genetic/epigenetic differences have been identified between normal and precancer (or cancer) cells (Phase 1), suggesting difficulty in targeting in this phase of carcinogenesis. The nature of the final step from precancer to cancer (Phase 2) remains unclear. However, the lung cancer rate rises as the 4th power of duration of smoking in continuing smokers, then remains constant when smoking stops[20]: the sustained incidence in ex-smokers indicates that the final rate-limiting step is independent of mutagen exposure, suggesting it may be caused by something other than a mutation.

The Postulate model (1) does not apply to all cancer types. For example, it is apparent that the model does not fit to age-incidence patterns of Hodgkin lymphoma and testicular cancer: their age-incidence curves have peaks that are inconsistent with the Postulate, two peaks several decades apart for Hodgkin lymphoma, and a distinct peak around the 3rd and 4th decade of life for testicular cancer[4]. For these and other cancer types to which the Postulate model does not fit,

**Table 1 | Summary of the fits of Peto-Mack Postulate to population-level age-specific incidence rates of various cancer types in males and females**

| Cancer type | Best Fit Models' $k$ values[a] | | Percent in the Subpopulations with Higher Propensities[b] | | Ratio of the Subpopulation Sizes with Lower vs. Higher Propensities[b] | | Highest Propensity Subpopulation is Larger[c] | |
|---|---|---|---|---|---|---|---|---|
| | Female | Male | Female | Male | Female | Male | Female | Male |
| Brain | 7, **10**, 13 | 7, 10, **13** | 0.31 | 0.44 | 4.2 | 4.8 | • | • |
| Gallbladder | 10, **13** | **10**, 13 | 0.01 | 0.00 | 91.1 | High = 0 | | |
| Kidney | **7**, 10, 13 | **7**, 10, 13 | 0.20 | 0.00 | 22.7 | High = 0 | | |
| Leukemia | **10** | 7, **10**, 13 | 0.35 | 0.48 | 18.7 | 26.6 | • | • |
| Melanoma of skin | 4, **7**, 10 | **7**, 10, 13 | 1.11 | 0.53 | 4.6 | 36.1 | • | • |
| Non-Hodgkin Lymphoma (NHL) | 7, 10, **13** | 7, 10, **13** | 0.55 | 0.72 | 15.3 | 19.5 | • | • |
| Stomach | **7** | 7, **10**, 13 | 0.12 | 0.17 | 27.2 | 36.5 | | |
| Pancreas | 7, **10**, 13 | 10, **13** | 0.07 | 0.09 | 134 | 114 | | |
| Breast | 7, **10** | - | 3.71 | - | 11.5 | - | | - |

[a]The best-fit model is shown in bold, and any models with an AIC difference of 2.0 or less from the best model are also listed.

[b]"Subpopulations with Higher Propensities" refers to those who reaches the high constant incidence rates before age 45 years old. "Subpopulations with Lower Propensities" refers to those who reaches the high constant incidence rates at or after age 45 years old. "High = 0" indicates the population comprises subpopulations with lower propensities only.

[c]The dot indicates the cancer type for which the best fit model indicated a higher proportion for the subpopulation with the highest propensity than the proportion of the subpopulation with the second highest propensity.

modifications of the model or alternative models need to be considered to explain their age-incidence patterns.

The conjecture that there exists a critically rate-limiting event in carcinogenesis whose rate is constant for decades in an individual presents a compelling paradigm for understanding cancer risk and developing targeted prevention strategies focused on the critically rate-limiting final event.

## Methods

### Second kidney cancer incidence data in the contralateral kidney

Primary diagnoses of renal-cell carcinoma were extracted from the cancer registry data of the Surveillance, Epidemiology, and End Results (SEER) Program of the US National Cancer Institute, which included data from 1975 to 2021 from 17 SEER registries[32]. Diagnoses involving only left- or right-sided kidney (i.e., no bilateral cases) prior to 2016 were included as the first diagnoses: 2016 was selected to allow sufficient follow-up of at least 5 years until the end of 2021 for the second primary to develop. SEER 22[4] contains the largest number of SEER registries, 22, but we used SEER 17, that do not contain five registries with only 2020–2021 data, because the 2-year follow up of the five registries was not sufficient for our subsequent primary incidence analysis. The second primary diagnoses in the contralateral kidney that developed at least a year later than the first primaries were identified, excluding the second primaries on the same side as the first or with unknown laterality. This resulted in 140,566 cases of the first primary renal-cell carcinoma and 2316 cases of subsequent contralateral primary renal-cell carcinoma diagnosed at least a year later than the first primaries. The follow-up length from the first primary to the earliest of the second contralateral primary, death, or the end of 2021 was used in assessing the incidence of the second renal-cell carcinoma in the contralateral kidney. Nelson-Aalen estimates of cumulative hazard rates[33] were obtained by SAS statistical software[34], overall and stratified by the age at the first primary diagnosis (categorized into <40 years old, 40–49 years old, 50–59 years old, 60–69 years old, and 70 years old or older). The estimated cumulative hazard rates of the second renal-cell carcinoma in the contralateral kidney were plotted according to years since the first primary, using R statistical software[35].

### Population-level age-specific cancer incidence rates

The population-level incidence rates of various cancer types by 5-year age group and sex were downloaded from SEER*Explorer, an interactive website for the SEER Program[4]: self-reported sex was considered in the study design and data were reported stratified by sex. Because this is not a subsequent cancer follow-up analysis, we utilized the cross-sectional data from the 22 SEER registries that cover 47.9% of the US population. The following cancer types/sites were analyzed: brain; gallbladder; kidney; leukemia; melanoma of skin; non Hodgkin lymphoma (NHL); pancreas; stomach; and breast (female only). Eleven 5-year age groups were analyzed: 35–39; 40–44; 45–49; 50–54; 55–59; 60–64; 65–69; 70–74; 75–79; 80–84; and 85+ years old (for female breast cancer, the 11 age groups started at age 25 due to the higher incidence rates of this cancer type in younger groups than the other cancer type).

### Armitage-Dole Model of carcinogenesis

In the classical Armitage-Doll Model of multistage carcinogenesis[6], initially considered similarly by Nordling[5], the population-level cancer incidence rate (per individual per year) is expressed by:

$$\text{Incidence Rate} = C \times (p_1 Age) \times (p_2 Age) \cdots \times (p_{k-1} Age) \times p_k = C' Age^{k-1} \quad (2)$$

where $C$ is proportional to the count of relevant cells, and $p_j$ is the rate of acquiring $j$th driver mutation per year per cell in the population of interest. In the log-log plot, the age-incidence relationship of the classic Armitage-Doll Model (2) would be a straight line with a y-intercept of $\log C' = \log(C p_1 p_2 \cdots p_{k-1} p_k)$ and a slope of $(k-1)$ with respect to the logarithm of age in the $x$-axis. That is, the population-level incidence rate increases as a power of age with the exponent of $(k-1)$, agreeing with the accumulation requirement of $k$ driver mutations of the multistage carcinogenesis theory and hallmarks of cancer that suggest $k = 6$–8. The Peto-Mack Postulate model (1) has two phases of individual-level cancer incidence: its Phase 1 part is equal to the classic Armitage-Dole Model (2).

### Empirical fit of Peto-Mack Model to age-specific cancer incidence data

We considered the Peto-Mack Postulate model (1) with up to six subpopulations with distinct propensity levels (i.e., each individual $i$'s

propensity $P_i'$ in the Postulate model (1) belongs to one of up to six subpopulations) and examined its fits to the observed population-level age-specific incidence rates of each of the various cancer types by sex. Specifically, we initially considered $k = 7$, consistent with the Nordling's[5] and Armitage-Doll Model[6], with 3, 4, 5, or 6 subpopulations of propensity which were assumed to reach a high constant incidence rate, set at 1.0% (or 0.5% in one example for an illustrative purpose) per year at 3, 4, 5, or 6 different ages, (65, 75, 85 years old), (55, 65, 75, 85 years old), (45, 55, 65, 75, 85 years old), and (35, 45, 55, 65, 75, 85 years old), respectively. The subpopulations' proportions in the population in each of the 3-, 4-, 5-, and 6-subpopulation models, $(\hat{p}65, \hat{p}75, \hat{p}85)$ $(\hat{p}55, \hat{p}65, \hat{p}75, \hat{p}85)$, $(\hat{p}45, \hat{p}55, \hat{p}65, \hat{p}75, \hat{p}85)$, and $(\hat{p}35, \hat{p}45, \hat{p}55, \hat{p}65, \hat{p}75, \hat{p}85)$, were estimated by a linearly constrained optimization of the least square method, minimizing the sum of the squares of the logarithm of the observed rate minus the logarithm of the model-fitted rate, summed over the 11 5-year age groups, with the constraint that each subpopulation proportion must be between 0 and 1 and the sum of the subpopulation proportions must not exceed 1. The estimation was performed using the R function *constrOptim*[35], which by default employs the optimization method of Nelder and Mead. The best parsimoniously fit model was selected among the 3-, 4-, 5-, and 6-subpopulation models using Akaike's Information Criterion (AIC)[27].

The estimation of the subpopulations' proportions provides information on the composition of the propensity levels in a population for a given cancer type. This information allows inference on selective pressure and early-life/childhood-onset of the specific cancer type. For example, very small subpopulations that reach the high constant rate at younger (reproductive) ages would imply selective pressure against the inherited components of higher propensity. On the other hand, an appreciable subpopulation with a high propensity would imply the presence of an early-life/childhood-onset subset of the specific cancer type under consideration.

Further analyses considered the identical setting as above with the only difference being the value of $k$: $k = 4$, 10, and 13 were considered in addition to $k = 7$.

## Reporting summary

Further information on research design is available in the Nature Portfolio Reporting Summary linked to this article.

## Data availability

Data used in this paper were "Surveillance, Epidemiology, and End Results (SEER) Program SEER*Stat Database: Incidence - SEER Research Data, 8 &17 Registries, Nov 2023 Sub (2000–2021) - Linked To County Attributes - Time Dependent (1990–2022) Income/Rurality, 1969–2022 Counties, National Cancer Institute, DCCPS, Surveillance Research Program, released April 2024, based on the November 2023 submission" available from the US SEER registry (https://seer.cancer.gov/data/).

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

## Acknowledgements
This work is partially supported by the National Cancer Institute grant, P30CA21765 (PI: Dr. Charles Roberts) and the American Lebanese-Syrian Associated Charities (ALSAC). We would like to thank Dr. Greg Armstrong and Dr. Jose Miguel Martinez for valuable support for this work.

## Author contributions
Y.Y. conceived the idea of the reported work, performed the data analysis, and drafted/revised the manuscript. Q.L. extracted the data on the second renal-cell carcinoma incidence in the contralateral kidney after the first diagnosis from the SEER registry, calculated its cumulative hazard rate estimates, and drafted the associated method. Both authors reviewed and approved the final draft of the manuscript.

## Competing interests
The authors declare no competing interests.
