## [Peer Review File · Nature Communications]

Single Rate-limiting Event of Carcinogenesis

Corresponding Author: Professor Yutaka Yasui

Version 0:

Reviewer comments:

Reviewer #1

(Remarks to the Author)

This is an interesting manuscript which puts forward a conceptual framework for thinking about individual level cancer risk, building on the earlier postulate by Peto and Mack. The authors propose a two phase model in which the risk of cancer within an individual increases exponentially with age up to a point which is mostly driven by somatic evolution and clonal competition and then plateaus when the stem cell compartment becomes somewhat oligo clonal and fixed with a small number of dominant clones. They support this idea with SEER data showing constant incidence rates over time for second primary cancers, specifically in the contralateral breast and kidney, and argue that these plateaus contradict the standard multistage carcinogenesis model that predicts continuous age related risk increase. They introduce a model that incorporates individual specific propensity parameters which determine both the age at which cancer risk plateaus and the level of the risk. They then show that, by assuming a small number of subpopulations with differing propensities, their model can fit the observed age incidence curves for different cancers. The work is certainly ambitious and raises some thought provoking ideas. That said, I had a few questions and major concerns which would be helpful if the authors can clarify or expand on:

The piecewise structure of the model an exponential increase followed by a flat incidence rate is intriguing but feels somewhat heuristic. Can they explain more about why they believe this structure captures a biologically meaningful transition, as opposed to being an artefact of data smoothing or competing risk effects in older populations? have they considered more mechanistic modelling approaches, such as those based on clonal expansion or mutation accumulation with age?

The authors acknowledge that different combinations of model parameters such as the number of required hit, the plateau rate, the distribution of propensities can lead to similar fits. Given this, I want to know how they can select the biological conclusions drawn from the best fit models? For example, how do they know the inferred subpopulation proportions aren't simply a byproduct of overfitting?

One of the central claims is the plateaued incidence rates in contralateral organs which reflects a fundamental shift in individual risk and relies mostly on registry data. But how did they account for potential biases such as differences in surveillance intensity, prior treatment effects, or mortality from the first cancer? Is it possible that the observed plateaus reflect these external factors rather than an underlying biological mechanism?

The idea of a latent individual level propensity is interesting, but I am wondering how this might map onto measurable quantities? Is this meant to represent inherited genetic risk, environmental exposure, something polygenic, or a combination? And in practical terms, do they envision this being something we can estimate for individuals?

The model is largely interpretive, but I wonder whether it could also be used predictively. Have they considered testing its predictions in individual level datasets for example in, familial registries or long term follow up cohorts to see whether the proposed phase transitions align with real world observations?

The model appears to fit several cancer types quite well, though with different inferred parameters. Are there cancer types you would expect this framework not to apply to? For example, where stochastic events play a larger role than stem cell lineage history?

(Remarks to the Author)

This interesting paper should be published, but it needs substantial revision. It offers a plausible explanation of a long-standing paradox in cancer epidemiology: the roughly constant breast cancer incidence rate after the probands' age at first diagnosis seen in their contralateral breasts and in their twins.

The authors cite the extensive recent evidence that many cancer types arise from mutant clones that have grown competitively to occupy a substantial fraction of the stem cell compartment. They suggest that this can evolve to a steady state in which the number of precancer cells an individual harbours (those requiring a single stochastic event to become a growing malignant cancer) remains roughly constant. This can involve dynamic competition with non-malignant clones (Colom Nature 2021 doi:10.1038/s41586-021-03965-7 should be cited). These are referred to as ARFT (at risk for transformation) cells. The usual terminology is precancer cells. An individual's subsequent cancer incidence will then be constant if the rate at which precancer cells undergo this final event is constant (a basic assumption in the classical multi-stage model). The rate at which an individual's number of precancer cells and hence their cancer risk increases initially, so the age by which they attain this steady state will be largely determined by genetic predisposition (as well as non-genetic exposures) that accelerates or bypasses earlier stages in carcinogenesis.

The authors also show that breast cancer incidence is constant following DCIS diagnosis, and that cancer incidence in the contralateral kidney following renal cell cancer diagnosis is also roughly constant irrespective of age at the first diagnosis. They also show that the rising age-incidence curves of various other cancers can be modelled by fitting several "propensity classes" that reach this steady state at different ages. Under this model a lifelong continuing increase in the age-specific population incidence rate reflects the increasing proportion who have reached this steady state, not an increasing number of precancer cells in each individual. Their division of the population into discrete "propensity" classes is a convenient approximation of the polygenic continuum of risk for the purpose of simulation, and this should be acknowledged.

Lung cancer incidence rises as a power (about 4) of duration of smoking then remains roughly constant when smoking stops (Peto Br J Cancer 2012 doi:10.1038/bjc.2012.97). The lung cancer data suggest that the number of precancer cells continues to increase in smokers and remains constant in ex-smokers, so each individual's risk rises as a power of time then plateaus abruptly when they stop smoking.

For cervical cancer an increasing proportion of women have had an active HPV infection that created one or more persistent precancers. Population incidence within birth cohorts thus remains roughly constant after middle age when active infection is much rarer (Plummer Int J Cancer 2012: doi:10.1002/ijc.26250). These examples of constant incidence after a carcinogen is withdrawn should be mentioned. They suggest that the authors' hypothesis, that the number of precancer cells reaches a plateau at a genetically determined age, does not apply to these cancers.

The constant incidence in ex-smokers implies that the final step in lung carcinogenesis is not affected by smoking, and is therefore unlikely to be mutational. Many genetic and epigenetic differences have been identified between normal and precancer or stage 1 cancer cells, and also between stage 1 and metastatic cancers, but as far as I know the nature of the final step from precancer to stage 1 cancer remains unclear. This interesting and understudied point is not widely recognised and might be mentioned.

The simple Armitage and Doll (Br J Cancer 1954) multistage model suggests that 7 rate-limiting steps are required to explain why the incidence rate of many cancers increases as the 6th power of age. However, subsequent modelling (e.g. Moolgavkar and Knudson JNCI 1981 doi: 10.1093/jnci/66.6.1037) shows that models with fewer stages and intermediate clonal expansion make slightly different predictions that fit the data equally well, and can also account for cancers with very different age-distributions, notably breast cancer and childhood cancers. This paper shows that extreme genetic heterogeneity can halve the slope of population incidence on a log-log plot (fig 4D). The high observed slopes in log-log age incidence plots for many cancers may reflect exponential growth of partially transformed cells modified by their population dynamics (birth-death and competition between clones) as well as the number of rate-limiting steps. A further factor affecting the observed power of age is the lag between inception and detection of a growing cancer. From ages 30-75 years a power of age is statistically indistinguishable from a lower power of (age minus 5 years). The similarity between the number of stages in the simple Armitage-Doll multistage model (about 7) and the number of "hallmarks of cancer" (6-8) is thus coincidental. All this could be mentioned, but for the purposes of this paper the main point is that the initial incidence rate can be modelled as a power of age.

The discussion should be abbreviated. An important inference is that the total number of precancer (ARFT) cells in an individual's different stem-cell compartments reach a maximum (the "phase transition") at approximately at the same age. However, an abrupt transition from rising numbers of precancer cells to a plateau is a sensible simplification for modelling but is implausible. The claim that because the factors that determine propensity (i.e. age-specific cancer risk) are strongly correlated within families "inherited germline mutations" (presumably meaning highly penetrant genes such as BRCA1 and BRCA2) are more important than the large number of weak polygenic germline variants that dominate cancer susceptibility is a non-sequitur and not true, particularly for breast cancer. Variation in risk is due more to polygenic variation than to high-risk genes. It is reasonable to infer that individual susceptibility (called "propensity" in the paper), which is largely inherited but is also affected by other risk factors, determines the age by which the plateau in incidence is reached. The claim that "the corollary of this individual-level age-risk pattern is that the last "hit" of multistage carcinogenesis is the critically rate-limiting event" detracts from the paper's simple hypothesis. Preventing the final step would prevent cancer under any model, but so would preventing any earlier step. The title of the paper therefore seems inappropriate.

The text needs editing, eg in the first sentence following the abstract "Large body of evidence" should be "A large body of

evidence”, and “which together makes the cell to escape” should be “which enable the cell to escape”.

“Exponential” and “exponentially” should be deleted or altered in several places. They are used to describe incidence rates which are a power of age (a straight line on a log-log plot). An exponential is a straight line on a log-linear plot. For hormone-dependent cancers such as breast cancer the rate increases much less steeply after menopause (see fig 1 in Peto and Mack 2000 and fig 5 in Moolgavkar and Knudson JNCI 1981 doi: 10.1093/jnci/66.6.1037). Thus “unlike the well-known exponential increase of population-level incidence with age” (abstract) should be “unlike the continuing increase of population-level incidence with age”.

I am happy to be identified to the authors if they want to discuss these comments whether or not their paper is accepted.

Version 1:

Reviewer comments:

Reviewer #1

(Remarks to the Author)

The authors have adequately addressed my concerns, and the revised version is much clearer. I am happy to recommend the manuscript for publication in its current form.

Reviewer #2

(Remarks to the Author)

This hypothesis and the supporting evidence justify publication. However, the paper should clarify the relationship between long-standing multi-stage models of carcinogenesis and what is novel about this paper. The incidence rate of many cancers increases roughly as the 6th power of age. The simplest Armitage and Doll (1954) model explains this by 7 rate-limiting steps, each occurring at a constant rate with no clonal expansion at intermediate stages. Moolgavkar et al (1981) showed that a 2-stage model with proliferation of intermediate cells makes similar predictions, and that more complex incidence patterns, including childhood cancers such as retinoblastoma and the much lower rate of increase of breast cancer after menopause, can be fitted by their model with reasonable age-related assumptions about rates of cell proliferation in the relevant tissues. At that time it was thought that genetic variation makes a minor contribution to population rates for most types of cancer, but this early work shows that very different models can explain the various age-distributions of different cancers. The model-fitting of different cancers in this paper (Table 1) shows that a wide range of population incidence patterns that continue to rise with age but decline below linearity on a log-log plot can also be fitted by a combination of rates that rise as a power of age then plateau at different ages. (The statement that “the high constant rate in Peto-Mack Postulate is not an estimable parameter in our analytic framework” is true when modelling population incidence rates, but the constant rate is observed directly in the contralateral breast cancer rate.) Cancers that have a peak (including Hodgkin’s disease, testicular cancer and all childhood cancers) cannot be fitted. The accepted interpretation of such peaks is age-related differences in cell division, the extreme example being retinal cells, which don’t divide at all after infancy. The suggestion that the age-distributions of all cancers that do not exhibit a peak are due to a uniform initial power law with heterogeneity of propensity and hence the age at plateau is implausible, so the inferences about the distribution of susceptibility for different cancers based on Table 1 are dubious. The comment about Wilm’s tumour (which is embryonal) and adult kidney cancers (which arise in different cells) should be deleted.

There is now a great deal of epidemiological as well as molecular evidence on susceptibility to different cancers and the cells they arise in, particularly for breast cancer. The simplest polygenic model in which genetic and environmental components multiply the risk predicts >100-fold range of risk in the population (Pharoah et al 2002: DOI: 10.1038/ng853). The contribution of BRCA1 and BRCA2 to familial risk is minor even in young women (Peto JNCI 1999: doi: 10.1093/jnci/91.11.943), so this is a reasonable approximation for the combined effect of non-genetic familial factors and the large number of low-penetrance variants. The polygenic risk score (Kramer et al 2020: doi: 10.1016/j.ajhg.2020.09.001), family history and various non-genetic risk factors (Akdeniz 2019: doi: 10.1016/j.breast.2018.11.005) are predictive of contralateral breast cancer. This evidence that the constant contralateral breast cancer rate is an average of different rates should be mentioned.

Most models, like the one presented here, assume that the age-specific cancer incidence rate is proportional to the number of precancer cells multiplied by the average rate per cell of the final transition to cancer. A key assumption in previous models was that precancer cells are rare and continue to increase with increasing age. That is why Peto and Mack observed that the constant contralateral breast cancer rate seems inconsistent with conventional models of carcinogenesis. The assumption that precancer cells are rare is challenged by the recent discovery that in many tissues the proportion of the stem cell compartment occupied by clones with multiple cancer-associated mutations increases to a high level. Based on this discovery, the authors’ postulate is that each individual’s number of precancer cells rises to a plateau, resulting in cancer incidence that increases roughly as a power of time up to an age determined by their cancer risk (propensity) then remains constant. The new evidence in this paper is the constant contralateral cancer rate following first diagnosis of renal cancer and breast DCIS.

The observation that the lung cancer rate rises as the 4th power of duration of smoking in continuing smokers then remains constant when smoking stops has an important implication that should be mentioned. The fact that the rate does not fall in ex-smokers shows that potent mutagens do not affect the rate of the final rate-limiting step, suggesting that it is not a mutation.

The reference to the single rate-limiting event proposed by Green and Evan (ref 28) is unhelpful and confusing. The model proposed here is a multi-stage process in which the final transition is described as the single crucial rate-limiting event. Style: Phrases such as “Two lines of sound evidence call for further appraisal” and “the high-quality epidemiological data on breast cancer incidence” would be improved by deleting “sound” and “high-quality”.

In summary, the paper should concentrate on breast cancer, present the new evidence on contralateral cancers, and note that the recent evidence on mutations in many tissues and the age distributions of many other cancers are consistent with the same model. Any stronger conclusions, particularly on mechanisms, are not justified.

I have uploaded a paper (Parish 1986) that is difficult to access that the authors might be interested in. It shows that the polygenic model gives a good fit to cancer incidence rates in mice that have already developed 0, 1, 2, 3 etc tumours.

Responses to the reviewers' comments on the manuscript NCOMMS-25-15107-T

We would like to sincerely thank the reviewers and the editor for their time and thoughtful feedback on our manuscript. We have considered all comments carefully, addressed and incorporated them as described below. The manuscript has been revised substantially based on the valuable comments and suggestions provided and is now greatly strengthened.

Reviewer #1:

Q1.1: This is an interesting manuscript which puts forward a conceptual framework for thinking about individual level cancer risk, building on the earlier postulate by Peto and Mack. The authors propose a two phase model in which the risk of cancer within an individual increases exponentially with age up to a point which is mostly driven by somatic evolution and clonal competition and then plateaus when the stem cell compartment becomes somewhat oligo clonal and fixed with a small number of dominant clones. They support this idea with SEER data showing constant incidence rates over time for second primary cancers, specifically in the contralateral breast and kidney, and argue that these plateaus contradict the standard multistage carcinogenesis model that predicts continuous age related risk increase. They introduce a model that incorporates individual specific propensity parameters which determine both the age at which cancer risk plateaus and the level of the risk. They then show that, by assuming a small number of subpopulations with differing propensities, their model can fit the observed age incidence curves for different cancers. The work is certainly ambitious and raises some thought provoking ideas.

R1.1: Thank you for your positive and encouraging comments.

Q1.2: That said, I had a few questions and major concerns which would be helpful if the authors can clarify or expand on:

The piecewise structure of the model an exponential increase followed by a flat incidence rate is intriguing but feels somewhat heuristic. Can they explain more about why they believe this structure captures a biologically meaningful transition, as opposed to being an artefact of data smoothing or competing risk effects in older populations? have they considered more mechanistic modelling approaches, such as those based on clonal expansion or mutation accumulation with age?

R1.2: Thank you for raising this question on the increase-to-plateau transition, a key feature of Peto-Mack Postulate, initially proposed based on a body of epidemiological evidence on breast cancer incidence in twins, relatives, and contralateral breast. We believe this key feature of the Postulate captures a biologically meaningful transition and explain our reasoning below.

1) **Biological meaning of the transition and its evidence**

We had described the hypothesized biological mechanism underlying the increase-to-plateau transition in the section of "Somatic evolution/cell competition effect" with supporting evidence. However, we fully agree with the reviewer, and are grateful for the reviewer's suggestion, that the transition's biological meaning would be clarified and strengthened if more mechanistic modeling approaches support it. Mitchell et al.²⁰ conducted an informative, well-designed cell-competition simulation whose results indeed demonstrated the abrupt transition, namely, a drastic and sudden reduction in clonal diversity with increased individual clone sizes as a result

of somatic evolution and cell competition, consistent with their empirical data. We have revised the section as follows.

“Accumulation of somatic mutations over time gives stem cells varying fitness advantages, driving cell competition among them, resulting in expansions of higher fitness clones and removal of the others.^{1-3,20,21} However, this process does not appear to persist throughout lifetime. As shown by Mitchell et al.²⁰ for hematopoiesis and reviewed by van Neerven and Vermeulen²¹ for various tissues, emerging evidence indicates a drastic shift in somatic evolution and cell competition with ageing, marked by a significant and sudden reduction in clonal diversity with increased individual clone sizes in stem-cell compartments. This sudden shift has also been demonstrated in an elegant mechanistic-modeling simulation of cell competition over ageing, using approximate Bayesian computation.²⁰ Consequently, a stem-cell compartment in ageing comprises a small number of clones, or even a single clone (“fixation”), reaching a static state with respect to the composition of the compartment’s cell population. For example, Mitchell et al. reported a reduction of 3-4 orders of magnitude in the number of stem cells contributing to hematopoiesis, from 20,000-200,000 to less than 20, by 75 years of age.²⁰ This bounded nature of somatic evolution/cell competition in ageing, whether it is due to spatial limitations^{20,21,25} or ageing-related factors,^{20,21} resonates with the two phases of individual-level cancer risk and the increase-to-plateau transition of Peto-Mack Postulate.”

2) Considerations of possible alternative explanations

Data smoothing

Had the Peto-Mack Postulate been based solely on modeling of cancer registry’s population-based age-specific incidence data, data smoothing or related issues of model (over)fitting may be of concern. This is because the registry’s age-specific incidence data are in a greatly aggregated form of individual risks, and models of different forms of individual risks could fit well with sufficient parameters/flexibility. What distinguishes Peto-Mack Postulate is its derivation based on a wide range of epidemiological data on cancer incidence including those in twins, relatives, and contralateral breast. Our manuscript amplifies evidence using the incidence in the contralateral kidney after renal-cell carcinoma. Thus, data smoothing of registry-based age-specific incidence data does not explain the increase-to-plateau transition.

Competing risk

Competing risk (i.e., mortality risk) directly influences the cumulative incidence (probability), but not the incidence rate, of the disease. Our analysis of the empirical fit of the Postulate model focused on incidence rates of various cancer types. Specifically, the estimation of incidence rates terminates the at-risk period for the disease at death, and the estimated rate quantifies the force of developing the disease per unit time at risk for the disease. The higher risk of death in older ages would lower the cumulative incidence of cancer because of less chance to develop cancer with higher risk of death. But the force of developing cancer while alive is not directly influenced by higher risk of death, as it is a quantity measured during the time a subject is alive and at risk for developing cancer. Thus, competing risk does not explain the transition of incidence rates. Similarly, the cumulative hazard rates of renal-cell carcinoma in the contralateral kidney are also unaffected by competing risk, being also a rate-based quantity.

In very old ages (e.g., in the 10th decade of life) when the mortality risk (i.e., competing risk) is very high and rapidly increasing, estimated incidence rates of a disease may be lower if the disease risk is correlated with the mortality risk. But this should only affect the very old ages (as seen in Figure 1).

Q1.3: The authors acknowledge that different combinations of model parameters such as the number of required hit, the plateau rate, the distribution of propensities can lead to similar fits. Given this, I want sure how they can select the biological conclusions drawn from the best fit models? For example, how do they know the inferred subpopulation proportions aren't simply a byproduct of overfitting?

R1.3: As the reviewer pointed out, models with different combinations of parameters led to similar fits. What is critical with this result is that the features of subpopulation proportions described using Table 1 were remarkably similar across the models with different combinations of parameters, as illustrated with Figure 4. In other words, while the evidence for a particular combination of model parameters is not strong, the Postulate model fits similarly well across a variety of parameter combinations with a set of common features of modeling results consistent with selection pressure and childhood/adolescent cancer risk. The stability of inferred subpopulation ratios across models with different parameterizations strengthens confidence in the conclusions' biological plausibility rather than overfitting artifacts. As model parsimony was achieved using AIC, balancing with goodness of fit, excessive overfitting was avoided in each model fitting. These points are now discussed along with Figure 4:

“These features are consistent across all panels, and their agreement across models with different parameterizations supports their biological plausibility, mitigating concerns about overfitting artifacts of the modeling.”

We also gave further discussion on related points with Figure S1 in Supplementary Information.

Q1.4: One of the central claims is the plateaued incidence rates in contralateral organs which reflects a fundamental shift in individual risk and relies mostly on registry data. But how did they account for potential biases such as differences in surveillance intensity, prior treatment effects, or mortality from the first cancer? Is it possible that the observed plateaus reflect these external factors rather than an underlying biological mechanism?

R1.4: Thank you for asking this question on one of our central claims. The reviewer pointed out that the registry data on the contralateral organ may be subject to bias such as surveillance intensity, treatment of the first cancer, or mortality from the first cancer. We explain below why we believe each of these sources of bias is an unlikely explanation of the constant incidence rate in the contralateral organ.

Surveillance intensity

Surveillance is naturally intensified by the diagnosis of the first primary cancer, and indeed could potentially make a rate that increases with time appear as flat. However, for a rate that increases as a power of age, the surveillance intensity would have to decrease drastically with age, to the extent that surveillance's ability to detect cancer decreases with the corresponding power of age. Such a drastic change in cancer detection due to surveillance intensity is implausible.

Treatment of the first cancer

Surgical treatment of the first cancer should not affect the incidence rate in the contralateral organ appreciably. Even if radiotherapy and/or chemotherapies of the first cancer influence the incidence rate in the contralateral organ, for a rate that increases as a power of age in the contralateral organ to appear constant, these treatments must increase the incidence rate in the contralateral organ drastically initially, followed by a gradual but extreme amount of decrease in

their risk elevation over time, to the extent that negate the power-law increase of the rate, which is unlikely. On the other hand, if these treatments decrease the incidence rate in the contralateral organ, their risk lowering effect must be small initially and become stronger with time to negate the drastic increase of the rate with age, which is also highly unlikely.

Mortality from the first cancer

As we described in R1.1 2) above, the competing risk of death should not affect the rate estimation and should not plateau the rate, except possibly in the very old ages.

Q1.5: The idea of a latent individual level propensity is interesting, but I am wondering how this might map onto measurable quantities? Is this meant to represent inherited genetic risk, environmental exposure, something polygenic, or a combination? And in practical terms, do they envision this being something we can estimate for individuals?

R1.5: As the reviewer pointed out, the concept of propensity encompasses inherited genetics, environmental/lifestyle factors shared among family members, and their complex interplay. Being a latent quantity, we cannot measure the propensity for an individual: we can only estimate its mean for a group of individuals which is meaningful if they share the same propensity. While currently unmeasurable at the individual level, propensity may eventually be approximated through integrating genetic, epigenetic, and environmental/lifestyle factors. We describe this in Discussion as follows:

“For propensity P_i to dictate the age at phase transition that “varies between families,” the factors that determine propensity P_i must be strongly correlated within families. That is, inherited germline mutations (instead of de novo germline mutations) and environmental/lifestyle factors that are shared strongly among family members must influence propensity P_i .”

Q1.6: The model is largely interpretive, but I wonder whether it could also be used predictively. Have they consider testing its predictions in individual level datasets for example in, familial registries or long term follow up cohorts to see whether the proposed phase transitions align with real world observations?

R1.6: Thank you for raising this point. To validate the model as suggested by the reviewer, we need a large enough group of individuals known to share the same level of propensity that is high enough to produce a sufficient number of cancer cases in the group so that the shape of the incidence rate in the group can be estimated. This is similar to the data that led Peto and Mack to their derivation of the Postulate (and the twin data in Ref. #10), although in their data families were aggregated. Specifically, they used incidence data in twins and 1st- and 2nd-degree relatives of breast cancer cases (i.e., individuals with similar propensity levels) and observed constant incidence rates among them after the first cancer. We appreciate the reviewer’s thoughtful suggestion and believe it may be possible if there is a large family registry with high-risk families, each with a sufficient number of 1st- (and 2nd-) degree relatives among whom the incidence is high enough so that the shape of the incidence rate among them can be estimated.

Q1.7: The model appears to fit several cancer types quite well, though with different inferred parameters. Are there cancer types you would expect this framework not to apply to? For

example, where stochastic events play a larger role than stem cell lineage history?

R1.7: Thank you for raising this important question. In terms of fitting the cancer registries' age-specific incidence rates, Hodgkin lymphoma, for example, has a bimodal age-incidence curve with two peaks separated several decades apart. Testicular cancer has a unimodal age-incidence curve with a peak around 3rd and 4th decades of life. The model fit would be different for these cancers and the estimated proportions of subpopulations would not generally increase with the subpopulation's age at the phase transition. We have added the following statement in the Discussion.

“The Postulate model (1) does not apply to certain types of cancer such as Hodgkin lymphoma, which has two peaks in its age-incidence curve several decades apart, and testicular cancer, which has a distinct peak around the 3rd and 4th decade of life. Modifications of the model are required to explain cancer incidence rates of these cancer types.”

We would like to thank Reviewer #1 for the time, thoughtful review, and constructive critiques.

Reviewer #2:

Q2.1: This interesting paper should be published, but it needs substantial revision. It offers a plausible explanation of a long-standing paradox in cancer epidemiology: the roughly constant breast cancer incidence rate after the probands' age at first diagnosis seen in their contralateral breasts and in their twins. The authors cite the extensive recent evidence that many cancer types arise from mutant clones that have grown competitively to occupy a substantial fraction of the stem cell compartment. They suggest that this can evolve to a steady state in which the number of precancer cells an individual harbours (those requiring a single stochastic event to become a growing malignant cancer) remains roughly constant. This can involve dynamic competition with non-malignant clones (Colom Nature 2021 doi:10.1038/s41586-021-03965-7 should be cited).

R2.1: Thank you for your positive and encouraging comments and pointing out our omission of the paper. We agree that the omitted paper is important and highly relevant, and is now cited in the Discussion.

Q2.2: These are referred to as ARFT (at risk for transformation) cells. The usual terminology is precancer cells.

R2.2: Thank you for pointing this out. We apologize for the terminology. We now use “precancer cells” throughout the manuscript.

Q2.3: Their division of the population into discrete “propensity” classes is a convenient approximation of the polygenic continuum of risk for the purpose of simulation, and this should be acknowledged.

R2.3: We agree with the reviewer's suggestion and have added to “Empirical fit of the Postulate model” the following statement:

“Note that between-individual heterogeneity in propensity, as represented in the Postulate model (1), exists on a continuum and extends beyond the discrete subpopulations, but we used the six subpopulations for modeling convenience here.”

Because the determinants of propensity include shared familial environmental/lifestyle factors as well as inherited genetics (including polygenic effects), as stated in the description of the propensity, we used “a continuum” instead of “polygenic continuum.”

Q2.4: Lung cancer incidence rises as a power (about 4) of duration of smoking then remains roughly constant when smoking stops (Peto Br J Cancer 2012). The lung cancer data suggest that the number of precancer cells continues to increase in smokers and remains constant in ex-smokers, so each individual’s risk rises as a power of time then plateaus abruptly when they stop smoking.

For cervical cancer an increasing proportion of women have had an active HPV infection that created one or more persistent precancers. Population incidence within birth cohorts thus remains roughly constant after middle age when active infection is much rarer (Plummer Int J Cancer 2012). These examples of constant incidence after a carcinogen is withdrawn should be mentioned. They suggest that the authors’ hypothesis, that the number of precancer cells reaches a plateau at a genetically determined age, does not apply to these cancers.

R2.4: We are grateful to the reviewer for pointing out these important examples of the plateaued incidence. We have incorporated them in Introduction as follows:

“There are other pieces of evidence for the plateaued incidence rates of cancer. For example, lung cancer incidence rises as the 4th power of the duration of smoking exposure then remains roughly constant when smoking stops (Peto Br J Cancer 2012). Incidence rates of cervical cancer which is influenced by HPV infection remain roughly constant within birth cohorts after middle age when active infection is much rarer (Plummer Int J Cancer 2012).”

We recognize that the reviewer is not only pointing out the incidence plateau of these cancer types, but also suggesting that the hypothesis, that the number of precancer cells reaches a plateau at a genetically determined age, does not apply to these cancers. Note, however, that our concept of propensity determinants did include environmental and lifestyle factors shared within families. As such, we incorporated the reviewer’s suggestion by considering the exposure to, and duration of, these environmental/lifestyle factors to influence the accumulation of driver mutations (i.e., propensity) and, consequently, the age at the phase transition. Specifically, we now state smoking and HPV exposures where variation of individual-level risk within and between subpopulations are discussed:

“For example, smoking habits²³ and HPV infection patterns²⁴ are major risk factors of lung cancer and cervical cancer, respectively, perhaps with between-individual variation within families that influence the propensity. Withdrawal of these major risk factors such as smoking cessation would significantly alter the propensity and the power-law increase of the number of precancer cells associated with the exposure.”

Q2.5: Many genetic and epigenetic differences have been identified between normal and precancer or stage 1 cancer cells, and also between stage 1 and metastatic cancers, but as far as I know the nature of the final step from precancer to stage 1 cancer remains unclear. This

interesting and understudied point is not widely recognised and might be mentioned.

R2.5: Thank you for pointing this out. We have incorporated the suggested point in Discussion where we focus on the final step as follows:

“Furthermore, many genetic/epigenetic differences have been identified between normal and precancer (or cancer) cells (Phase 1) The nature of the final step from precancer to cancer (Phase 2) remains unclear, however.”

Q2.6: The simple Armitage and Doll (Br J Cancer 1954) multistage model suggests that 7 rate-limiting steps are required to explain why the incidence rate of many cancers increases as the 6th power of age. However, subsequent modelling (e.g. Moolgavkar and Knudson JNCI 1981 doi: 10.1093/jnci/66.6.1037) shows that models with fewer stages and intermediate clonal expansion make slightly different predictions that fit the data equally well, and can also account for cancers with very different age-distributions, notably breast cancer and childhood cancers. This paper shows that extreme genetic heterogeneity can halve the slope of population incidence on a log-log plot (fig 4D). The high observed slopes in log-log age incidence plots for many cancers may reflect exponential growth of partially transformed cells modified by their population dynamics (birth-death and competition between clones) as well as the number of rate-limiting steps. A further factor affecting the observed power of age is the lag between inception and detection of a growing cancer. From ages 30-75 years a power of age is statistically indistinguishable from a lower power of (age minus 5 years). The similarity between the number of stages in the simple Armitage-Doll multistage model (about 7) and the number of “hallmarks of cancer” (6-8) is thus coincidental. All this could be mentioned, but for the purposes of this paper the main point is that the initial incidence rate can be modelled as a power of age.

R2.6: While many researchers, including Moolgavkar and colleagues, have elegantly refined the Armitage-Doll framework of modeling registry-based age-specific incidence data, we focus here on the implications on individual-level risk derived from Peto and Mack’s empirical findings and postulate, which originated from twin/family-based data and contralateral breast data rather than registry-based age-specific incidence rates. Thus, we did not discuss the modeling work of cancer registries’ age-specific incidence rates in detail. Nonetheless, we fully acknowledge the importance of this line of research and have now included a reference to Moolgavkar and Knudson as suggested, along with Armitage and Doll.

Q2.7: The discussion should be abbreviated.

R2.7: To shorten the Discussion significantly, we have moved the long discussion on the variation of individual-level risk within and between subpopulations and Figure 5 (now Figure S1) to Supplementary Information.

Q2.8: An important inference is that the total number of precancer (ARFT) cells in an individual’s different stem-cell compartments reach a maximum (the “phase transition”) at approximately at the same age. However, an abrupt transition from rising numbers of precancer cells to a plateau is a sensible simplification for modelling but is implausible.

R2.8: Regarding the abruptness of the increase-to-plateau transition, we now describe its evidence as part of the hypothesized biological mechanism underlying the transition: the section

of “Somatic evolution/cell competition effect” has been revised with its supporting evidence. Specifically, an informative well-designed cell-competition simulation is now cited whose results indeed demonstrated the abrupt transition, namely, a drastic and sudden reduction in clonal diversity with increased individual clone sizes as a result of somatic evolution and cell competition, consistent with the corresponding empirical data. The revised text now reads:

“As shown by Mitchell et al.²⁰ for hematopoiesis and reviewed by van Neerven and Vermeulen²¹ for various tissues, emerging evidence indicates a drastic shift in somatic evolution and cell competition with ageing, marked by a significant and sudden reduction in clonal diversity with increased individual clone sizes in stem-cell compartments. This sudden shift has also been demonstrated in an elegant mechanistic-modeling simulation of cell competition over ageing, using approximate Bayesian computation.²⁰”

Q2.9: The claim that because the factors that determine propensity (i.e. age-specific cancer risk) are strongly correlated within families “inherited germline mutations” (presumably meaning highly penetrant genes such as BRCA1 and BRCA2) are more important than the large number of weak polygenic germline variants that dominate cancer susceptibility is a non-sequitur and not true, particularly for breast cancer.

R2.9: Our intent of the Discussion section in question was to state that factors determining propensity appear strongly correlated within families due to inherited genetics, familial environmental/lifestyle factors, and their combinations. We removed the reference to “weak polygenic germline variants” to focus on this intent. The revised text now reads:

“inherited germline mutations (instead of de novo germline mutations) and environmental/lifestyle factors that are shared strongly among family members must influence propensity P_i .”

Q2.10: It is reasonable to infer that individual susceptibility (called “propensity” in the paper), which is largely inherited but is also affected by other risk factors, determines the age by which the plateau in incidence is reached. The claim that “the corollary of this individual-level age-risk pattern is that the last “hit” of multistage carcinogenesis is the critically rate-limiting event” detracts from the paper’s simple hypothesis. Preventing the final step would prevent cancer under any model, but so would preventing any earlier step. The title of the paper therefore seems inappropriate.

R2.10: We agree with the reviewer that the “corollary of this individual-level age-risk pattern” statement was not carefully worded. But we respectfully view the last ‘hit’ somewhat differently from the reviewer’s characterization. The abundance of precancer cells with various sets of driver mutations suggests that precancer cells can develop in a number of different ways and at high frequencies. Thus, preventing the development of such common precancer cells seems difficult. On the other hand, the constant incidence period where precancer cells go through the final step to transform to cancer is long and much less frequent (based on the comparison of frequencies of precancer cells vs. cancer incidence), but the majority of the cancer cases come from the constant-rate phase. Based on the reviewer’s comment, we have removed the “corollary of this individual-level age-risk pattern” statement and the revised text now reads:

“Given the constant incidence, which signifies a state requiring only of a single stochastic “hit” for malignant transformation,^{14,15} and the abundance of precancer cells relative the rarity of

cancer,¹⁻⁸ we reasoned that the final step for transforming from precancer to cancer represents the most biologically restrictive step in the multistage carcinogenesis process, i.e., the critically rate-limiting event.”

We therefore respectfully maintain that the title remains aligned with the central message we aim to convey in the manuscript.

Q2.11: The text needs editing, eg in the first sentence following the abstract “Large body of evidence” should be “A large body of evidence”, and “which together makes the cell to escape” should be “which enable the cell to escape”.

R2.11: We thank the reviewer for pointing these out. The manuscript, including the errors pointed out by the reviewer here, have now been reviewed and extensively edited/corrected.

Q2.12: “Exponential” and “exponentially” should be deleted or altered in several places. They are used to describe incidence rates which are a power of age (a straight line on a log-log plot). An exponential is a straight line on a log-linear plot. For hormone-dependent cancers such as breast cancer the rate increases much less steeply after menopause (see fig 1 in Peto and Mack 2000 and fig 5 in Moolgavkar and Knudson JNCI 1981 doi: 10.1093/jnci/66.6.1037). Thus “unlike the well-known exponential increase of population-level incidence with age” (abstract) should be “unlike the continuing increase of population-level incidence with age”.

R2.12: Thank you for pointing this out and we apologize for our careless use of “exponential.” We now distinguish “exponential increase with age” (a straight line on a log-linear plot against age) and “power-law increase with age” (a straight line on a log-log plot against age) and use the latter and associated expressions (e.g., “increase as a power of age”) throughout the manuscript.

Q2.13: I am happy to be identified to the authors if they want to discuss these comments whether or not their paper is accepted.

R2.13: We would like to thank Dr. Peto for your time, thoughtful review, and constructive critiques. We are especially grateful for your offer of the opportunity to discuss your comments whether or not our manuscript is accepted. We would like to take you up on your offer to discuss further after a decision is made by the journal’s editor.

Responses to the reviewers' comments on the manuscript NCOMMS-25-15107-A

We would like to thank Reviewer #2 and the editor for their time and thoughtful feedback on our revised manuscript. We have considered all comments carefully, addressed and incorporated them as described below.

Q1. This hypothesis and the supporting evidence justify publication. However, the paper should clarify the relationship between long-standing multi-stage models of carcinogenesis and what is novel about this paper. ... this early work shows that very different models can explain the various age-distributions of different cancers. The model-fitting of different cancers in this paper (Table 1) shows that a wide range of population incidence patterns that continue to rise with age but decline below linearity on a log-log plot can also be fitted by a combination of rates that rise as a power of age then plateau at different ages.

R1. Thank you for this suggestion. We apologize for failing to clarify the relationship between long-standing multi-stage models of carcinogenesis and what is novel about this paper. We have now included the following paragraph at the beginning of Discussion to address the issues raised by the reviewer.

“Mathematical modeling of population-level age-specific cancer incidence has a distinguished history. Nordling⁵ and Armitage and Doll⁶ demonstrated that population-level mortality (incidence) rates of many types of cancer rise as the 6th power of age, implying seven rate-limiting steps. Moolgavkar and Knudson²⁸ proposed an elegant clonal-expansion model with only two stages, estimating parameters representing cell kinetics of normal and intermediate cells. Pike et al.²⁹ and Rosner and Colditz³⁰ advanced the Armitage-Doll framework for breast cancer by introducing “breast tissue age,” reflecting a life-course sequence of hormonal and reproductive events known to modify the risk. Despite their conceptual differences, these landmark models all fit population-level incidence well. Our work continues this tradition while offering a fundamentally different perspective, the Peto-Mack Postulate (Figure 3), which Peto and Mack formulated from breast cancer findings¹⁵ unavailable to earlier models. After further supporting the Postulate with contralateral organ incidence data (DCIS and renal cell carcinoma), we showed that the Postulate’s two-phase structure, i.e., an initial Armitage-Doll-like rise in incidence as a power of age, followed by a sustained high plateau, provides a novel and robust fit to population-level age-specific incidence rates across a variety of cancer types (Table 1). This fit was enabled by individual “propensity,” which has familial and hereditary components and determines both the individual’s age-dependent rate (rising as a power of age) and the age at which the plateau begins.”

Q2. The statement that “the high constant rate in Peto-Mack Postulate is not an estimable parameter in our analytic framework” is true when modelling population incidence rates, but the constant rate is observed directly in the contralateral breast cancer rate.

R2. We have clarified this point by modifying the statement as follows:

“the high constant rate in Peto-Mack Postulate is not an estimable parameter in our analytic framework from the population-level age-specific incidence rates (but the constant rate is observed directly in the contralateral breast cancer rate¹⁵).”

Q3. Cancers that have a peak (including Hodgkin's disease, testicular cancer and all childhood cancers) cannot be fitted. The accepted interpretation of such peaks is age-related differences in cell division, the extreme example being retinal cells, which don't divide at all after infancy. The suggestion that the age-distributions of all cancers that do not exhibit a peak are due to a uniform initial power law with heterogeneity of propensity and hence the age at plateau is implausible, so the inferences about the distribution of susceptibility for different cancers based on Table 1 are dubious. The comment about Wilm's tumour (which is embryonal) and adult kidney cancers (which arise in different cells) should be deleted.

R3. We certainly did not intend to suggest that the Postulate model fit well to age-distributions of all cancer types that do not exhibit a peak. In response to Reviewer #1's earlier question, "Are there cancer types you would expect this framework not to apply to?", we listed Hodgkin lymphoma and testicular cancer as apparent examples for which the model does not apply due to their peaks in age-incidence curves. We now modified the paragraph to avoid this unintended suggestion as follows and focused on the fact that the Postulate model does not apply to all cancer types.

"The Postulate model (1) does not apply to all cancer types. For example, it is apparent that the model does not fit to age-incidence patterns of Hodgkin lymphoma and testicular cancer: their age-incidence curves have peaks that are inconsistent with the Postulate, two peaks several decades apart for Hodgkin lymphoma and a distinct peak around the 3rd and 4th decade of life for testicular cancer.⁴ For these and other cancer types to which the Postulate model does not fit, modifications of the model or alternative models need to be considered to explain their age-incidence patterns."

Regarding the Wilms' tumor, the SEER registry combines Wilms' tumor and adult kidney cancers as 'kidney cancer' which led to our earlier phrasing. We have now removed it as suggested.

Q4. There is now a great deal of epidemiological as well as molecular evidence on susceptibility to different cancers and the cells they arise in, particularly for breast cancer. The simplest polygenic model in which genetic and environmental components multiply the risk predicts >100-fold range of risk in the population (Pharoah et al 2002: DOI: 10.1038/ng853). The contribution of BRCA1 and BRCA2 to familial risk is minor even in young women (Peto JNCI 1999: doi: 10.1093/jnci/91.11.943), so this is a reasonable approximation for the combined effect of non-genetic familial factors and the large number of low-penetrance variants. The polygenic risk score (Kramer et al 2020: doi: 10.1016/j.ajhg.2020.09.001), family history and various non-genetic risk factors (Akdeniz 2019: doi: 10.1016/j.breast.2018.11.005) are predictive of contralateral breast cancer. This evidence that the constant contralateral breast cancer rate is an average of different rates should be mentioned.

R4. We have now added/modified the following sentences, referencing Kramer et al. and Akdeniz et al. as the reviewer suggested, to emphasize the point the reviewer raised.

"Contralateral breast cancer incidence rates are associated with both genetic and environmental/lifestyle risk factors.^{25,26} These observations are consistent with the Postulate

framework, in which heterogeneity in propensity reflects the combined influence of multiple genetic and environmental/lifestyle determinants.”

“Note that between-individual heterogeneity in propensity, as represented in the Postulate model (1), exists on a continuum due to genetic and environmental/lifestyle risk factors.^{25,26”}

Q5. Most models, like the one presented here, assume that the age-specific cancer incidence rate is proportional to the number of precancer cells multiplied by the average rate per cell of the final transition to cancer. A key assumption in previous models was that precancer cells are rare and continue to increase with increasing age. That is why Peto and Mack observed that the constant contralateral breast cancer rate seems inconsistent with conventional models of carcinogenesis. The assumption that precancer cells are rare is challenged by the recent discovery that in many tissues the proportion of the stem cell compartment occupied by clones with multiple cancer-associated mutations increases to a high level. Based on this discovery, the authors' postulate is that each individual's number of precancer cells rises to a plateau, resulting in cancer incidence that increases roughly as a power of time up to an age determined by their cancer risk (propensity) then remains constant. The new evidence in this paper is the constant contralateral cancer rate following first diagnosis of renal cancer and breast DCIS.

R5. We thank the reviewer for clearly stating this important point. This important perspective is now highlighted in the revised Discussion as described in **R1**, while we avoided conclusions on mechanisms in the revision, in line with the reviewer's guidance in **Q9**, by using expressions throughout the manuscript such as “hypothesized” and “potential.”

Q6. The observation that the lung cancer rate rises as the 4th power of duration of smoking in continuing smokers then remains constant when smoking stops has an important implication that should be mentioned. The fact that the rate does not fall in ex-smokers shows that potent mutagens do not affect the rate of the final rate-limiting step, suggesting that it is not a mutation.

R6. We would like to thank the reviewer for emphasizing this observation and its implication. We have now added the following in Discussion:

“However, the lung cancer rate rises as the 4th power of duration of smoking in continuing smokers then remains constant when smoking stops:²⁰ the sustained incidence in ex-smokers indicates that the final rate-limiting step is independent of mutagen exposure, suggesting that it may be caused by something other than a mutation.”

Q7. The reference to the single rate-limiting event proposed by Green and Evan (ref 28) is unhelpful and confusing. The model proposed here is a multi-stage process in which the final transition is described as the single crucial rate-limiting event.

R7. Following the reviewer's comment, we have now removed the paragraph referring to Green and Evan's Cooperative Hypothesis of carcinogenesis which we believe is not necessarily suggesting that the carcinogenesis is a single-stage process, but we did not want to confuse readers.

Q8. Style: Phrases such as “Two lines of sound evidence call for further appraisal” and “the high-quality epidemiological data on breast cancer incidence” would be improved by deleting “sound” and “high-quality”.

R8. Thank you for this suggestion. We have removed these adjectives.

Q9. In summary, the paper should concentrate on breast cancer, present the new evidence on contralateral cancers, and note that the recent evidence on mutations in many tissues and the age distributions of many other cancers are consistent with the same model. Any stronger conclusions, particularly on mechanisms, are not justified.

R9. In line with the reviewer’s guidance, our manuscript has been revised to have the initial focus on breast cancer (the origin of the Peto-Mack Postulate), highlights the new evidence on contralateral cancers supporting the Postulate, and shows the Postulate model’s goodness of fit to age-incidence patterns of a variety of cancer types, while avoiding conclusions on mechanisms. We have added terms such as “hypothesized” and “potential” to avoid any impression that we are making a conclusion on mechanisms.

Q10. I have uploaded a paper (Parish 1986) that is difficult to access that the authors might be interested in. It shows that the polygenic model gives a good fit to cancer incidence rates in mice that have already developed 0, 1, 2, 3 etc tumours.

R10. We would like to thank the reviewer for kindly providing this paper which we read with great interest.